# Gastrostomy and congenital anomalies: a European population-based study

Ester Garne ,[1] Joachim Tan,[2] Maria Loane,[3] Silvia Baldacci ,[4] Elisa Ballardini,[5] Joanne Brigden,[2] Clara Cavero-Carbonell,[6] Laura García-Villodre,[6] Mika Gissler,[7,8] Joanne Given,[3] Anna Heino,[8] Sue Jordan,[9] Elizabeth Limb,[2] Amanda Julie Neville,[10] Anke Rissmann ,[11] Michele Santoro,[4] Ieuan Scanlon,[9] Stine Kjaer Urhoj ,[1,12] Diana G Wellesley,[13] Joan Morris [2]

## ABSTRACT

**Objective** To report and compare the proportion of children with and without congenital anomalies undergoing gastrostomy for tube feeding in their first 5 years.

**Methods** A European, population-based data-linkage cohort study (EUROlinkCAT). Children up to 5 years of age registered in nine EUROCAT registries (national and regional) in six countries and children without congenital anomalies (reference children) living in the same geographical areas were included. Data on hospitalisation and surgical procedures for all children were obtained by electronic linkage to hospital databases.

**Results** The study included 91 504 EUROCAT children and 1 960 272 reference children. Overall, 1200 (1.3%, 95% CI 1.2% to 1.6%) EUROCAT children and 374 (0.016%, 95% CI 0.009% to 0.026%) reference children had a surgical code for gastrostomy within the first 5 years of life. There were geographical variations across Europe with higher rates in Northern Europe compared with Southern Europe. Around one in four children with Cornelia de Lange syndrome and Wolf-Hirschhorn syndrome had a gastrostomy. Among children with structural anomalies, those with oesophageal atresia had the highest proportion of gastrostomy (15.9%).

**Conclusions** This study including almost 2 million reference children in Europe found that only 0.016% of these children had a surgery code for gastrostomy before age 5 years. The children with congenital anomalies were on average 80 times more likely to need a gastrostomy before age 5 years than children without congenital anomalies. More than two-thirds of gastrostomy procedures performed within the first 5 years of life were in children with congenital anomalies.

## INTRODUCTION

Gastrostomy is used for infants and children with major, long-term feeding problems and failure to thrive.[1] The procedure of inserting a permanent gastrostomy is generally considered safe.[2] The main indications for a permanent or long-term gastrostomy are severe neurological disorders and congenital anomalies associated with severe feeding problems.[3–5] A 19% increase in the number of surgical procedures for gastrostomy on children over the years 1997–2009

## WHAT IS ALREADY KNOWN ON THIS TOPIC

⇒ Gastrostomy for tube feeding is undertaken for children with major feeding problems and failure to thrive, but estimates of its prevalence are not readily available.

⇒ The main indications for gastrostomy in children are severe neurological disorders and congenital anomalies with severe feeding problems.

## WHAT THIS STUDY ADDS

⇒ Only 16 per 100 000 children with no congenital anomalies have gastrostomy surgery within the first 5 years of life.

⇒ Children with congenital anomalies are on average 80 times more likely to need a gastrostomy before age 5 years than children without congenital anomalies.

⇒ More than two-thirds of gastrostomy surgical procedures before age 5 years are performed in children with major congenital anomalies.

## HOW THIS STUDY MIGHT AFFECT RESEARCH, PRACTICE AND/OR POLICY

⇒ The existing heterogeneity in prevalence of surgery for gastrostomy in young children across Europe should be explored in future research.

⇒ Risks from this study of surgery for gastrostomy in children with specific congenital anomalies can be provided to their parents at the time of diagnosis.

has been reported using national data from the USA.[6] It has also been shown that, after surgery for gastrostomy, weight-for-length z-scores increase,[7] as does the quality of life for the child and the caregiver.[8] A UK study found a point prevalence of gastrostomy in situ in January 2020 of 84 per 100 000 children for the age group 0–19 years and 79.6 per 100 000 for the younger age group 0–4 years.[5] Little is known about the frequency of gastrostomy in children with specific congenital anomalies that are associated with feeding problems and failure to thrive.

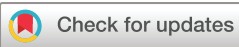

For numbered affiliations see end of article.

**Correspondence to**
Dr Ester Garne; egarne@dadlnet.dk

**Table 1** Number of children with and without congenital anomalies and proportions with a surgery code for gastrostomy before age 5 years, by registry

| | EUROCAT children | | | Reference children | | |
|---|---|---|---|---|---|---|
| | Number of children included in study | Children with surgery for a feeding tube age 0–4 years | | Number of children included in study | Children with surgery for a feeding tube age 0–4 years | |
| | | Number | Percentage (95% CI) | | Number | Percentage (95% CI) |
| Denmark, Funen 1995–2014 | 2423 | 40 | 1.8 (1.3 to 2.4) | 100 748 | 33 | 0.03 (0.02 to 0.05) |
| Finland 1997–2014 | 38 324 | 552 | 1.5 (1.4 to 1.7) | 911 679 | 255 | 0.03 (0.03 to 0.03) |
| Italy, Tuscany 2005–2014 | 4225 | 35 | 0.9 (0.6 to 1.2) | 23503* | 0 | 0.00 (—) |
| Italy, Emilia Romagna 2008–2014 | 5381 | 51 | 1.0 (0.8 to 1.4) | 223 995 | 21 | 0.01 (0.01 to 0.02) |
| Spain, Valencian Region 2010–2014 | 4260 | 30 | 0.8 (0.6 to 1.2) | 168 563 | 12 | 0.01 (0.00 to 0.01) |
| UK, Wales 1998–2014 | 17 448 | 210 | 1.3 (1.2 to 1.5) | 531 784 | 53 | 0.01 (0.01 to 0.02) |
| UK, Thames Valley 2005–2013 | 3845 | 46 | 1.3 (1.0 to 1.8) | n/a | | |
| UK, E Midlands and S Yorkshire 2003–2012 | 11 278 | 158 | 1.6 (1.3 to 1.8) | n/a | | |
| UK, Wessex 2004–2014 | 4320 | 78 | 2.1 (1.7 to 2.6) | n/a | | |
| Total | 91 504 | 1200 | 1.3 (1.2 to 1.6)† | 1 960 272 | 374 | 0.016 (0.009 to 0.026)† |

*Reference children Tuscany 10% of population.
†Estimated percentages and 95% CIs are calculated using a random-effects meta-analysis of individual registries' estimates and not from the total numbers given in the final row of the table.
n/a, not available.

The aims of this EUROlinkCAT Study were to report (1) the frequency of surgery for gastrostomy in children with congenital anomalies up to the age of 5 years compared with the general population of all children without congenital anomalies living in the same geographical areas and (2) the prevalence of gastrostomy in children with specific congenital anomalies.

## METHODS

This is a European, population-based data-linkage cohort study using routinely collected health service data. The study includes data from nine European surveillance of congenital anomalies (EUROCAT) registries (national and regional) in six countries (see table 1). Children with major congenital anomalies as defined by EUROCAT[9] born between 1995 (or the first year of the EUROCAT registry if later) and 2014 were included (EUROCAT children). For five registries, data on all liveborn children without congenital anomalies from the same population covered by the registry and the same birth years were included as the reference population (reference children). The Tuscany registry used a 10% random sample of their population as the reference children. There were no reference children available from the three English registries.

Data on hospitalisation and surgical procedures for all children up to the child's 10th birthday or end of 2015, whichever came earlier, were obtained by electronic linkage to the hospital databases used in the regions and countries. This ensured that at least 1 year of follow-up after birth was available for each child. We included children up to 5 years of age in this study as the number of children reaching the age of 10 years with the specific congenital anomalies under consideration was limited. Details of the methods used in the EUROlinkCAT Study including the linkage methods have been published elsewhere.[10 11]

The hospital databases in Finland, Denmark (Funen), Tuscany in Italy and England (East Midlands & South Yorkshire, Thames Valley and Wessex) covered hospitalisation

of the study population in the whole country. For Wales, this included procedures carried out in England. For Valencian Region in Spain and Emilia Romagna in Italy, the hospital databases covered the same region as the EUROCAT registry.

Surgical procedures were coded according to the coding systems used in the national health systems. Italy and Spain used ICD-9-CM (the International Classification of Diseases, ninth revision-Clinical Modification) for the study period, England and Wales used OPCS-4 (Classification of Interventions and Procedures), and Finland and Denmark used national adaptions of NCSP (NOMESCO Classification of Surgical Procedures). The surgery codes used for gastrostomy were 43.11 and 43.19 in ICD-9-CM, G341 and G342 in OPCS-4.8, and JDB in NCSP.

Data were analysed for all children with major congenital anomalies and additional analyses performed on children with the following specific anomalies: spina bifida, hydrocephaly, congenital heart defects and severe congenital heart defects, cleft lip with and without cleft palate, cleft palate, oesophageal atresia, Down syndrome, Goldenhar association, Di George syndrome, Noonan syndrome, Williams syndrome, Cri du chat syndrome, Wolf-Hirschhorn syndrome, Angelmann syndrome and Cornelia de Lange syndrome. These anomalies all had specified ICD-9 and ICD-10 codes in the EUROCAT database and were selected from the EUROlinkCAT list of anomaly subgroups[10] by a paediatrician (EG) as associated with feeding problems.

There was no parent or public involvement in the study.

## Statistical methods

Each registry standardised their data using the EUROlinkCAT common data model which enabled them to run a centrally written analysis script on their individual case data. The proportion of children receiving a surgical gastrostomy code before the age of 5 years was calculated using Kaplan-Meier Survival Analysis (KM) to account for the censoring occurring on 1 January 2016. This enabled children who were born between 2010 and 2014 and who did not reach age 5 years to be included in the analysis. Each registry submitted the numbers of births, the numbers of children with a gastrostomy code before the age of 5 years and the KM estimates, by type of congenital anomaly, to the central results repository at Ulster University via a secure portal (no individual case data were provided to Ulster University). These results for all registries were then provided to the researchers via the same portal. The CIs for the KM survival analysis estimates were calculated by STATA using the $\ln(-\ln(S(t)))$ transformation. To obtain pooled estimates of the percentage of children with surgical gastrostomy code across registries, random-effects inverse-variance meta-analyses were performed using the $\ln(-\ln(S(t)))$ transformation. For several anomalies where no children in a single registry received a surgical gastrostomy code, the upper 95% confidence limit (CI) was calculated

using the exact binomial estimate, the percentage of children having gastrostomy was estimated to be 0.001% and the lower 95% CI was calculated assuming symmetry on the $\ln(-\ln(S(t)))$ scale. The meta-analysis was performed in Stata (V.15).

## RESULTS

The study included 91 504 EUROCAT children with major congenital anomalies and 1 960 272 reference children. Overall, 1200 (1.3%, 95% CI 1.2% to 1.6%) EUROCAT children and 374 (0.016%, 95% CI 0.009% to 0.026%) reference children had a surgery code for gastrostomy within the first 5 years of life. There were geographical differences across Europe with higher rates in the Nordic countries and UK (1.3%–2.1% of EUROCAT children) and lower rates in Italy and Spain (0.8%–1.0% of EUROCAT children). In the five registries with reference children for the whole population (Funen, Finland, Wales, Emilia Romagna and Valencian Region), there were 883 EUROCAT and 374 reference children with a surgery code for gastrostomy, in other words 70% of all gastrostomies within the first 5 years were performed on children with congenital anomalies.

Table 2 describes the number of children with specific congenital anomalies and the proportions with surgery codes for gastrostomy. Around one in four of the children with Wolf-Hirschhorn syndrome or Cornelia de Lange syndrome had a surgery code for gastrostomy before the age of 5 years, although the 95% CIs are wide due to the small number of children diagnosed with these rare syndromes. For children with oesophageal atresia, the proportion was 15.9% (95% CI 13.5% to 18.5%). The larger group of children with congenital heart defects had a lower prevalence of surgery codes for gastrostomy (0.5%, 95% CI 0.4% to 0.7%) and for children with severe congenital heart defects, the proportion was 1.3% (95% CI 0.8% to 2.0%). For children with facial clefts, the percentage with a surgery code for gastrostomy was highest for children with cleft palate (4.1%, 95% CI 3.4% to 4.9%).

## DISCUSSION

This study with European population-based data showed that surgery for gastrostomy within the first 5 years of life was on average 80 times more frequent in children with congenital anomalies than in children without congenital anomalies from the same geographical areas. The study also showed a difference between Northern and Southern Europe: children with congenital anomalies were overall twice as likely to receive gastrostomy in the Nordic countries and UK when compared with Italy and Spain; this could be due in part to the regional coverage of hospital databases in Valencian Region and Emilia Romagna. There are limited data published on the frequency of gastrostomy in children with specific

**Table 2** Meta-analysis of proportion of children with specific congenital anomalies having a surgery code for gastrostomy before 5 years of age

| Anomaly | Number of liveborn children | Children with a surgery for gastrostomy at age 0–4 years (adjusted for censoring) | |
|---|---|---|---|
| | | Number | Percentage (95% CI) |
| Spina bifida | 591 | 11 | 2.1 (1.0 to 3.7) |
| Hydrocephaly | 1082 | 52 | 5.5 (3.3 to 8.4) |
| Congenital heart defects | 26 442 | 125 | 0.5 (0.4 to 0.7) |
| Severe congenital heart defects | 5693 | 72 | 1.3 (0.8 to 2.0) |
| Cleft lip with or without cleft palate | 2961 | 37 | 1.4 (1.0 to 2.0) |
| Cleft palate | 2879 | 109 | 4.1 (3.4 to 4.9) |
| Oesophageal atresia | 915 | 138 | 15.9 (13.5 to 18.5) |
| Down syndrome | 3640 | 62 | 1.1 (0.5 to 2.2) |
| Goldenhar association | 301 | 27 | 9.7 (6.5 to 13.6) |
| Di George syndrome | 292 | 25 | 7.8 (3.3 to 14.9) |
| Noonan syndrome | 154 | 12 | 8.2 (3.6 to 15.1) |
| William syndrome | 115 | 4 | 2.9 (0.4 to 10.7) |
| Cri du chat syndrome | 57 | 8 | 15.8 (6.0 to 29.8) |
| Wolf-Hirschhorn syndrome | 37 | 8 | 23.9 (3.4 to 54.6) |
| Angelmann syndrome | 44 | 2 | 3.3 (0.2 to 15.9) |
| Cornelia de Lange syndrome | 34 | 9 | 29.2 (11.8 to 49.2) |

congenital anomalies and this study contributes to this literature.

A study from Germany showed that 24% of newborns in 2009–2013 with oesophageal atresia had a gastrostomy during their neonatal hospital stay.[12] This is higher than the 15.9% found in our study, where the proportion was a calculated average for nine regions in Europe and also included children born in an earlier time period.

The proportion of children with congenital heart defects having surgery for gastrostomy was low at 0.5%, but the risk was still 30 times higher than in the reference population (0.016%). For children with severe congenital heart defects, gastrostomy for feeding after neonatal cardiac surgery improves growth velocity in their first year of life.[13]

Despite the multicentre dataset, case numbers were too small to analyse by registry and over time. A study analysing data over time in the whole paediatric population in the USA showed that the number of procedures for gastrostomy increased every year from 1997 to 2009.[6] Also studies from Australia and Sweden have shown increase over time in the insertion and use of permanent feeding tubes in the paediatric population.[14 15] It is likely that surgeries for gastrostomy also increased in the geographical areas included in our study during the 20-year study period. A European study on children with cerebral palsy also found a difference in prevalence of gastrostomy across Europe with the highest prevalence in Sweden and the lowest prevalence in Portugal.[16] The study also found a low prevalence in Northern England. Our study did not find a low prevalence of gastrostomy in the English registries. Furthermore, our study cannot say

why there are differences in the use of gastrostomy. These differences may be cultural in nature, with different levels of acceptance by parents, or regional or national, such as access to the surgery and doctors' practices when discussing the surgery for gastrostomy with parents.

We do not have data on the duration of the tube feeding through the gastrostomy and if the gastrostomy was removed as soon as the feeding problems were resolved. It should be acknowledged that both the presence of feeding problems at a certain age and the surgery for gastrostomy to treat them will increase the stress for the child and the family.[17]

The main strength of this study is that the population-based data included all children with major congenital anomalies and not only those referred to tertiary hospitals. Further, the EUROCAT registries have high levels of case ascertainment and use standardised definitions and coding of congenital anomalies to ensure consistency across Europe. The data in the hospital databases were also standardised as part of the EUROlinkCAT Study before analysis scripts were written and distributed. The study also has limitations. The percentage of children with a surgery code for gastrostomy may be higher than presented here as the surgery codes may not always be registered in the hospital databases and variations in coding practices may mean that some relevant codes were not identified. There may be problems with the registration of procedures before the newborns have their permanent name or ID number allowing them to be matched to their surgery data, and some children may have been referred for surgery outside the area covered by the hospital databases in this study (although this

affected only two registries). All of these would lead to an underestimation of the prevalence of gastrostomy in the study population.

## CONCLUSIONS

This study including almost 2 million reference children in Europe found that only 0.016% of these children had a surgery code for gastrostomy before age 5 years. The children with congenital anomalies were on average 80 times more likely to need a gastrostomy before age 5 years than the children without congenital anomalies. This is important information when counselling the parents about the expected morbidity for their child after a prenatal or postnatal diagnosis of a specific congenital anomaly associated with feeding problems.

**Author affiliations**
[1]Department of Paediatrics and Adolescent Medicine, Lillebaelt Hospital-University Hospital of Southern Denmark, Kolding, Denmark
[2]Population Health Research Institute, St George's, University of London, London, UK
[3]Centre for Maternal, Fetal and Infant Research, INHR, Ulster University, Newtownabbey, UK
[4]Unit of Epidemiology of Rare Diseases and Congenital Anomalies, Institute of Clinical Physiology, National Research Council, Pisa, Italy
[5]Neonatal Intensive Care Unit, Paediatric Section, IMER Registry (Emilia Romagna Registry of Birth Defects), Department of Medical Sciences, University of Ferrara, Ferrara, Italy
[6]Rare Diseases Research Unit, Foundation for the Promotion of Health and Biomedical Research in the Valencian Region (FISABIO), Valencia, Spain
[7]Department of Molecular Medicine and Surgery, Karolinska Institutet, Stockholm, Sweden
[8]Knowledge Brokers, Finnish Institute for Health and Welfare, Helsinki, Finland
[9]Faculty of Medicine, Health and Life Science, Swansea University, Swansea, Wales, UK
[10]Azienda Ospedaliero-Universitaria di Ferrara, Registro IMER, Ferrara, Italy
[11]Malformation Monitoring Centre Saxony-Anhalt, Medical Faculty Otto-von-Guericke University, Magdeburg, Germany
[12]Department of Public Health, University of Copenhagen, Copenhagen, Denmark
[13]Wessex Clinical Genetics Service, Princess Anne Hospital, Southampton, UK

**Contributors** JM, ML and EG designed the EUROlinkCAT Study and obtained funding. CC-C, MG, AJN, DGW and EG contributed data from their EUROCAT registries. SKU, JT, AH, MS, LG-V, SB, EB, CC-C, IS, SJ and JB linked the EUROCAT data to the hospital databases. ML and JG performed the standardisation of the variables in collaboration with the local registries. AR and EG defined the surgical variables. JM and EL performed the statistical analysis. EG wrote the manuscript. All authors reviewed and revised the manuscript. EG is acting as guarantor.

**Funding** This project has received funding from the European Union's Horizon 2020 research and innovation programme under grant agreement no. 733001.

**Competing interests** None declared.

**Patient and public involvement** Patients and/or the public were not involved in the design, or conduct, or reporting, or dissemination plans of this research.

**Patient consent for publication** Not required.

**Ethics approval** All EUROCAT registries obtained ethical, governance and other permissions for the data linkage according to their national legislation and arrangements. University of Ulster obtained ethics permission for the Central Results Repository on 15 September 2017 (Institute of Nursing and Health Research Ethics Filter Committee, number FCNUR-17-000).

**Provenance and peer review** Not commissioned; externally peer reviewed.

**Data availability statement** Data are available on request.

**ORCID iDs**
Ester Garne http://orcid.org/0000-0003-0430-2594
Silvia Baldacci http://orcid.org/0000-0002-7626-1202
Anke Rissmann http://orcid.org/0000-0002-9437-2790
Stine Kjaer Urhoj http://orcid.org/0000-0002-2069-9723
Joan Morris http://orcid.org/0000-0002-7164-612X

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
