## [Reviewer comments · BMJ Paediatrics Open]

This paper was submitted to a another journal from Archives of Disease in Childhood but declined for publication following peer review. The authors addressed the reviewers' comments and submitted the revised paper to BMJ Paediatrics Open. The paper was subsequently accepted for publication at BMJ Paediatrics Open.

ARTICLE DETAILS

TITLE (PROVISIONAL)	Gastrostomy and congenital anomalies - a European population-based study
AUTHORS	Garne, Ester Tan, Joachim Loane, Maria Baldacci, Silvia Ballardini, Elisa Brigden, Joanne Cavero-Carbonell, Clara García-Villodre, Laura Gissler, Mika given, joanne Heino, Anna Jordan, Sue Limb, Elizabeth Neville, Amanda Julie issmann, Anke Santoro, Michele Scanlon, leuan Urhoj, Stine Kjaer Wellesley, Diana G Morris, Joan

VERSION 1 – REVIEW

REVIEWER	Reviewer name: Mr. Mark Powis Institution and Country: Leeds General Infirmary, United Kingdom of Great Britain and Northern Ireland Competing interests: None
REVIEW RETURNED	20-Mar-2022

GENERAL COMMENTS	I enjoyed reading the paper, but am unsure what it offers the clinician other than a breakdown of gastrostomy incidence by syndrome. I am unsure whether this information will affect their practice, but might be useful for the societies that are associated with the syndromes and parents. Being a surgeon I am surprised by the high incidence of gastrostomy inpatients with oesophageal atresia. Typically only those patients with long-gap or pure atresia get a gastrostomy unless there are associated issues with the VACTERL association. I wonder how many patients in this group just had oesophageal atresia and how many had other issues? Given that reference data comes from only 5 registries and three of these are in areas where there is a low incidence of gastrostomy
--

	placement, do the authors feel that their reference percentage is actually an underestimate? With respect to coding. This is often one of the worst things done at many hospitals. Were the patients coded for gastrostomy alone or combined with other procedures such as fundoplication for instance? Depending on your search you could have again potentially underestimated the children having gastrostomies sited.
--	---

REVIEWER	Reviewer name: Dr. Peter Flom Institution and Country: Peter Flom Consulting New York, United States Competing interests: None
REVIEW RETURNED	03-May-2022

GENERAL COMMENTS	I confine my remarks to statistical aspects of this paper. These were well done, and I recommend publication.
---